# Anti-Thrombogenicity Study of a Covalently-Attached Monolayer on Stent-Grade Stainless Steel

**DOI:** 10.3390/ma14092342

**Published:** 2021-04-30

**Authors:** Tairan Yang, Brian De La Franier, Michael Thompson

**Affiliations:** Department of Chemistry, University of Toronto, 80 St. George Street, Toronto, ON M5S 3H6, Canada; tairan.yang@mail.utoronto.ca (T.Y.); brian.delafranier@mail.utoronto.ca (B.D.L.F.)

**Keywords:** anti-thrombogenic surface, stent-grade stainless steel, blood exposure

## Abstract

Implantable devices fabricated from austenitic type 316L stainless steel have been employed significantly in medicine, principally because the material displays excellent mechanical characteristics and corrosion resistance. It is well known, however, that interaction of exposure of such a material to blood can initiate platelet adhesion and blood coagulation, leading to a harmful medical condition. In order to prevent undesirable surface platelet adhesion on biomaterials employed in procedures such as renal dialysis, we developed an ultrathin anti-thrombogenic covalently attached monolayer based on monoethylene glycol silane chemistry. This functions by forming an interstitial hydration layer which displays restricted mobility in the prevention of surface fouling. In the present work, the promising anti-thrombogenic properties of this film are examined with respect to platelet aggregation on 316L austenitic stainless steel exposed to whole human blood. Prior to exposure with blood, all major surface modification steps were examined by X-ray photoelectron spectroscopic analysis and surface free-angle measurement by contact angle goniometry. End-stage anti-thrombogenicity detection after 20 min of blood exposure at 100 s^−1^, 300 s^−1^, 600 s^−1^, 750 s^−1^, and 900 s^−1^ shear rates revealed that a significant reduction (>90%) of platelet adhesion and aggregation was achieved for surface-modified steel, compared with untreated material. This result is confirmed by experiments conducted in real time for 60-minute exposure to blood at 100 s^−1^, 600 s^−1^, and 900 s^−1^ shear rates.

## 1. Introduction

Coronary artery disease (CAD) is a debilitating condition where a buildup of plaque forms which causes the occurrence of arterial narrowing (stenosis) [1]. Thus far, a significant number of solutions have been developed to prevent stenosis from occurring, including coronary bypass surgery, balloon angioplasty, as well as coronary stents [2]. In particular, the poor biocompatibility of bare metal stents (BMS) has prevented the successful integration of this implant amongst all patients suffering from stenosis. Upon implantation of a bare metal stent, the artery undergoes re-narrowing, known as restenosis. Intimal hyperplasia, thrombus formation and cell injury during percutaneous transluminal coronary angioplasty (PTCA) are all factors contributing to the occurrence of restenosis [3,4,5].

Thrombus formation occurs when medical implants or devices are in contact with whole blood such that different components of the blood such as proteins and cells adsorb onto the surface of the implant [6]. It is very common for aggregates of platelets and fibrin to form thrombi on the surface of the coronary stent [6]. Without adequate options to treat the thrombus formation, this can diminish the effects of the stent implant such that re-stenosis can occur. Stent thrombosis can often lead to acute myocardial infarction and death. A vast array of approaches have been proposed to overcome both re-stenosis and thrombosis through functionalization of the surface of the coronary stent using various modifications and coatings. These include the employment of coatings via anti-proliferative drug molecules such as paclitaxel or rapamycin to fabricate the so-called drug-eluting stent (DES) [7,8,9,10,11], self-assembling molecular systems [12], especially peptide amphiphiles [13,14], carbon nanodeposit [15] and polymeric [16] systems, and re-endothelization via attachment of antibodies for endothelial cells [17,18].

The myriad of suggested approaches to prevent thrombus formation and re-stenosis do not necessarily display a high level of overall stability. By far the most common technology at the present time is the DES, as mentioned above. This technology, however, does not reduce the risk of developing thrombosis. Even with use of DES, anticlotting agents such as aspirin or clopidogrel are often administered, since clot formation can potentially affect how well the drug is eluted from the stent surface [19]. In fact, a concern associated with DES is that patients with these implants are at an increased risk of developing late stent thrombosis (~30 days to 1 year upon implantation of the stent) [19,20]. Moreover, the components of DES layers can be cytotoxic [21].

More recent advances in drug eluting stents include heparin-eluting stents [22,23,24,25] and combined heparin–nitric-oxide-eluting stents [26,27], which elute the anti-coagulant drug heparin and anti-bacterial nitric oxide. Several different methods are used to measure the amount of platelet adhesion and thrombus formation to these coatings, but these only typically reduce thrombus formation to a level in the range of 60–80%. As such, heparin-eluting stents still result in significant platelet adhesion and thrombus formation, in addition to potentially causing side effects such as heparin-induced thrombocytopenia [28].

Other recent work has included phenolic-amine chemistry combined with bivalirudin, which is a thrombin inhibitor, in order to combat thrombogenesis [29]. This layer was only able to reduce platelet activation and adhesion by 60–70%, which is quite limited. These coatings are also difficult to apply and use several expensive chemicals in thick layers to achieve their desired effects.

Graphene-coated stents have also been investigated for their anti-thrombogenic properties on stainless steel [30,31,32]. These coatings may show greater reduction in platelet adhesion compared to heparin-eluting stents, with some reducing adhesion by 80–90%, but these also show slightly elevated cytotoxicity compared to bare stents. Anti-thrombogenic hydrogel coatings have also seen advances in recent years [33,34], which display reasonable reduction in platelet adhesion to stainless steel with slightly over 90% reduction in platelet adhesion observed.

Medical grade stainless steel is the most common material employed for stent fabrication [35]. The reasons for the use of this material are associated with its bulk metallic and surface properties. Materials required for an expendable stent must have sufficient mechanical plasticity to fit the required diameter size within a blood vessel when deployed. In addition, self-expanding stents must be manufactured from material that offers enough elasticity so that the stents can be compressed and expanded within an affected vessel while retaining high radial hoop strength to prevent the blood vessel from recoiling or collapsing once fully deployed [36]. In terms of surface chemistry, stainless steel provides exceptional anti-corrosion characteristics. The alloy is composed predominantly of iron, chromium, nickel, molybdenum, and manganese. The high chromium content (~17 wt.%) provides the alloy with superior corrosion resistance, as well as mechanical advantages in hardness and strength [37].

Several of the surface treatments mentioned above have been used to modify the surface of 316L stainless steel in attempts to enhance its hemocompatibility properties. In the present work, we demonstrate that an ultrathin monoethylene glycol (MEG-OH), which is covalently attached to the steel surface, is capable of a dramatic reduction in platelet aggregation from blood. Such an effect has been previously shown with regard to polymeric biomaterials conventionally employed for conduits in renal dialysis and bypass surgery [38,39]. This result is attributed to the instigation of an interstitial water layer with reduced mobility properties [40,41].

This monoethylene glycol coating is robust, easy to apply, low-cost, and has shown a much higher degree of anti-thrombogenicity when applied to plastic polymers, compared to many drug-eluting stents [38,39]. As such, it is believed that this coating will also provide excellent anti-thrombogenicity when applied to 316L stainless steel, and will do so in a robust and low-cost manner. This could be used as a simple way to reduce thrombogenicity in stent implants, and improve biocompatibility of these devices. With this work, we aim to determine the ability of this coating to reduce thrombogenesis on 316L stainless steel as a step towards its use in implantable stents.

## 2. Materials and Methods

### 2.1. Reagents and Materials

All glassware involved in the silanization process was pre-treated with 5% (*v/v*) octadecyltrichlorosilane (OTS) in anhydrous toluene inside a glove box maintained under inert atmosphere. 2-(3-trichlorosilylpropyloxy) ethyl-trifluoroacetate (MEG-TFA) linker used for stainless steel surface silanization was synthesized as described previously [42]. All silanization was performed under inert (N_2_) and anhydrous (P_2_O_5_) atmosphere.

Concentrated sulfuric acid (98%) was purchased from ACP Chemicals Inc., Saint-Leonard, QC, Canada. Anhydrous toluene, ACS grade toluene, ethanol, dimethyl sulfoxide (DMSO), octadecyltrichlorosilane (OTS), and 98% 3,3′-dihexyloxacarbocyanine iodide (DiOC_6_) fluorescent dye were obtained from Sigma-Aldrich, Oakville, ONT, Canada. ACS grade chloroform, n-pentane, acetone, and methanol were purchased from Fisher Scientific, Ottawa, ONT, Canada. Annealed austenitic type 316L stainless steel foil, mirror-polished on both sides, with thickness of 2.0 mm, was purchased from Goodfellow Ltd., Cambridge, UK. Additionally, 1.0 mm thick, annealed, 2B mill polished on both sides, austenitic type 316L stainless steel plates (1.0 mm thick, annealed, 2B mill polished on both sides) were obtained from Stainless Supply, Ltd., Monroe, NC, USA.

Human blood was available from a single source through Canadian Blood Services in collaboration with the Krembil Research Institute of Western Hospital, Toronto, ONT, Canada. No individual donor was employed or associated with the research conducted and described in this paper.

### 2.2. Mechanical and Electrochemical Polishing of Steel Samples

No. 2B and buffed No. 8 (mirror-polished) finish are both achieved through mechanical polishing [43]. Electrochemical polishing of all steel samples was performed by Electro-Kleen Alloy Polishing Ltd., Mississauga, ON, Canada. The samples are submersed in an electrolytic solution, such as a mixture of H_2_SO_4_ and H_3_PO_4_. A direct electric current is then run through the electrolytic solution, making the stainless steel become the anode and a nearby metallic conductor the cathode in an electrolytic reaction. As the polarized film covers the indents and protrusions on the steel surface, the rate of metallic dissolution is higher at protrusion than at indents because of higher polarized film thickness and the resulting increased electric resistance at indents, ultimately leaving the steel surface microscopically featureless.

### 2.3. Surface Cleaning of Steel Samples

Steel samples were subjected to successive sonication in ACS grade n-pentane, ACS grade acetone, then 95% ethanol, for 20 min each. Afterwards, each sample was carefully rinsed with a large quantity of 95% ethanol, followed by a copious amount of distilled water. Cleaned stainless steel was subsequently dried under a steady stream of N_2_ gas. All dried steel samples were then individually immersed and oxidized in piranha solution as follows: 10 mm × 10 mm × 2 mm samples (for surface analysis) were placed within a test tube. Samples with dimensions of 50.8 mm × 25.4 mm × 1 mm (for blood flow experiments) were placed tilted against the glassware in a 120 mL beaker, which is capable of holding up to six samples. Each test tube or beaker was subsequently put in a hot water bath pre-heated to 90 °C. Piranha solution was prepared by adding three equivalents of 98% concentrated sulfuric acid to one equivalent 30% hydrogen peroxide. Approximately 2 mL of piranha solution was dispensed into each test tube, and about 100 mL of piranha solution was poured into the 120 mL beaker to fully immerse all steel samples. After 30 min, all oxidized samples were removed from the piranha solution. Each sample was then rinsed with a large amount of distilled water to wash off residual piranha solution on the surface, followed by 3 times the amount of ACS grade methanol. Next, samples were individually transferred to clean glassware and placed inside an oven maintained at 150 °C. After two hours, steel samples were taken out of the oven and left to cool down to room temperature for 15 min. Finally, each sample was plasma-cleaned using an expanded tabletop plasma cleaner with a maximum power of 30 V for 5 min. One oxidized sample was stored in a clean scintillation vial under nitrogen awaiting X-ray photoelectron spectroscopic (XPS) surface characterization.

### 2.4. Surface Silanization of Steel Samples

After plasma cleaning, all oxidized steel samples intended for surface modification were placed immediately inside a humidity chamber for 24 h to facilitate water vapor physisorption onto the steel surface. The humidity chamber was maintained by an aqueous solution of magnesium nitrate hexahydrate (Mg(NO_3_)_2_·6H_2_O) at relative humidity of 78% and 23 °C. Next, steel samples were transferred inside a glove box, along with MEG-TFA trichlorosilane linker stock solution, micropipettes, silanized glassware and Parafilm^®^. Next, 10 µL MEG-TFA linker was pipetted and diluted with 10 mL anhydrous toluene. Afterwards, 2.0 mL 1/1000 (*v/v*) MEG-TFA/anhydrous toluene solution was portioned into each silanized scintillation vial containing one 10 mm × 10 mm × 2.0 mm stainless steel sample. For 50.8 mm × 25.4 mm × 1.0 mm steel samples, a minimum of 20.0 mL 1/1000 (*v/v*) MEG-TFA/anhydrous toluene solution was required to fully immerse three samples of such dimension within a 150 mm diameter, 65 mm deep silanized Pyrex^®^ petri dish. The scintillation vials containing steel samples immersed in MEG-TFA/anhydrous toluene solution were screw-capped and subsequently sealed with Parafilm^®^. The petri dishes were placed in a desiccator before removing from the glove box. The scintillation vials and the desiccator were then placed on a spinning plate for two hours. When silanization was completed, each stainless-steel sample was rinsed three times with ACS grade toluene, followed by five minutes of sonication in ACS grade toluene. Then, the same rinsing and sonication process was repeated with ACS grade chloroform on each steel sample, followed by drying under nitrogen stream. One MEG-TFA modified stainless steel sample was stored under nitrogen in a clean scintillation vial, as previously discussed, for surface analysis.

For solvolysis of the trifluoroacetic acid (TFA) group, 10 mm × 10 mm × 2 mm MEG-TFA surface modified steel samples, placed in 20 mL clean scintillation vials, were individually immersed in 2.0 mL 1/1 (*v/v*) ethanol/distilled water on a spinning plate for overnight solvolysis at room temperature. Each 50.8 mm × 25.4 mm × 1.0 mm steel sample was placed in a Falcon^®^ 50 mL conical centrifuge tube, and the volume of 1/1 (*v/v*) ethanol/distilled water used for solvolysis was increased to 40 mL. All steel samples were subsequently dried under gentle nitrogen stream. One MEG-OH-modified steel sample was stored under nitrogen in a clean scintillation vial, as previously discussed, for XPS surface characterization, while others were stored for later blood perfusion experiments.

### 2.5. Surface Characterization

Select bare, oxidized, MEG-TFA-modified, and MEG-OH-modified 316L stainless steel samples were subjected to surface characterization by XPS. Angle-resolved XPS was analyzed using the Thermo Scientific AI Kα probe (Thermo Fisher Scientific, East Grinstead, UK). Take-off angles of 20° and 90° relative to surface were chosen to be analyzed for each sample. Peak fitting and data analysis of all XPS spectra, including survey scans and elemental narrow scans of C1s, O1s, Si2p, F1s, Cr2p, Fe2p, Mn2p, Mo3d, Ni2p, were processed using the accompanying Avantage^®^ and CasaXPS^®^ software, Casa Software Ltd., Teignmouth, UK. Binding energy calibration was performed on all spectra by shifting C1s peak to 284.8 eV prior to peak fitting and data analysis.

Surface wettability of all 316L stainless steel samples after each major step of surface modification was measured using CAM 101 optical tensiometer (contact angle meter), constructed by KSV Instruments Ltd., Monroe, CT, USA. Distilled water was consistently used as the testing liquid.

### 2.6. Parallel-Plate Flow Chamber Preparation and Assembly

A custom-made, single-pass, parallel-plate blood perfusion chamber was consistently used for whole human blood surface thrombogenicity analysis (Figure 1). A transparent 50.8 mm × 25.4 mm × 1.0 mm dimension plain glass coverslip was adapted from a commercial microscope slide. On each glass coverslip, two 1.0 mm diameter entrance and exit holes, 10.2 mm and 12.7 mm away from each edge, and 30.4 mm apart from each other, were drilled perpendicularly to the glass surface. Two threaded female Luer adaptors made from PMMA were mounted and glued directly upon the inlet and outlet holes with Henkel Loctite Hysol^®^ 0151 thixotropic epoxy paste adhesive (Henkel Adhesives, Mississauga, ON, Canada). Two threaded 1/16 inch, straight, male polyethylene tubing connectors, each connected with 75.0 cm and 150.0 cm long 1/16 inch/1/8-inch (inside/outside) diameter moisture-resistant polyethylene vacuum tubing, were ultimately screwed into the glued female adaptors. All Luer connections and insertions were then sealed with Parafilm^®^ extensively to prevent potential air or liquid leakage. The channel slide was made from clear, impact-resistant, both-sides-smooth, 0.0508 mm thick polycarbonate film, with up to 10% thickness tolerance. The polycarbonate film was cut into a size of 50.8 mm × 25.4 mm, and the flow channel was created by carving out a rectangular space with 4.0 mm width and 35.0 mm length.

The assembly of the blood perfusion chamber proceeded as follows. Both the glass coverslip and channel slide were cleaned and sterilized with 95% ethanol. The channel slide was then gently applied onto the coverslip, making sure the entrance and exit holes were unobstructed and completely exposed within the rectangular flow channel. Afterwards, a thin layer of distilled water was sprayed onto the channel slide to serve as lubricant and to prevent the formation of air bubbles. The assembly of blood perfusion chamber was completed by sandwiching the channel slide with a 50.8 mm × 25.4 mm × 1.0 mm bare or MEG-OH-surface-modified steel plate. Each side of the chamber was extensively sealed with Parafilm^®^ and further stabilized with small clamps.

### 2.7. Blood Flow Shear Rate and Regulation

In this study, shear rates of 100 s^−1^, 300 s^−1^, 600 s^−1^, 750 s^−1^, and 900 s^−1^ were chosen for whole human blood thrombogenicity analysis on bare and MEG-OH-surface-modified 316L stainless steel plates. The corresponding volumetric flow rate and required blood volume at each chosen shear rate were calculated [44] and displayed in Table 1, given that the channel width is 4.0 mm and the channel height is 0.0508 mm. A syringe pump utilizing pulling mode was used throughout all whole human blood thrombogenicity experiments to produce a uniform blood flow rate at all times.

### 2.8. Fluorescent Labelling of Platelets

3,3′-dihexyloxacarbocyanine iodide (DiOC_6_, λ_exc_/λ_em_ = 484/501 nm) fluorescent dye stock solution was prepared by dissolving 98% DiOC_6_ in ACS grade dimethyl sulfoxide (DMSO) at 1 mM concentration. The working solution was diluted to 10 µM concentration and stored in an aluminum-foil-wrapped vial and stored in a refrigerator. Collected whole human blood was stored in 6 mL lithium heparin Becton Dickinson plastic vacutainers. Whole human blood sample intended for 20 min surface thrombogenicity analysis was subsequently fluorescence-labeled with DiOC_6_ working solution at 1/100 (*v/v*) ratio. For 60 min surface thrombogenicity analysis, DiOC_6_ working solution was added to the whole human blood at 1/30 (*v/v*) ratio. The vacutainers containing fluorescence-labelled blood were then wrapped with aluminum foil and placed on a rotator for at least 20 min before use.

### 2.9. Equipment for Blood Perfusion Experiments

Pre-assembled flow chambers (5 bare and 5 MEG-OH-modified steel samples for each intended shear rate experiment) were prepared for surface thrombogenicity analysis against whole human blood under a BX61W1 Olympus, confocal microscope (Center Valley, PA, USA), equipped with an EMC2 Q-imaging CCD camera, Rolera, QImaging, (Advanced Imaging Concepts, Princeton, NJ, USA) The required exposure time was set to 300 ms without camera gain. The excitation laser source used was an arc mercury lamp, X-cite 120 PC fluorescence illumination system, EXFO, (Excelitas Technologies, Waltham, MA, USA). A green fluorescent filter cube set FITC 3540B Semrock, (IDEX Health & Science, Rochester, NY, USA). which combines a 35 nm wide excitation bandpass filter centered at 482 nm and a 40 nm emission bandpass filter centered at 536 nm, was used to filter fluorescent signals. The dimension of the camera chip used was 1004 × 1002 with pixel size of 8 × 8 µm, and the objective lens used was a 40× Plan N (Olympus) with a numerical aperture of 0.65. The microscope and camera setup allowed recording of live videos, from which specific frames were extracted. Extracted frames were then processed with imageJ^®^ software to calculate surface coverage due to platelet adhesion, aggregation, and thrombus formation on prepared substrates.

Before starting blood perfusion, the parallel-plate flow chamber was stabilized on the testing platform and fitted under the microscope, followed by a 2.0 mL phosphate-buffered saline injection to purge and test leak tightness. One-to-two drops of distilled water were added onto the coverslip for objective lens immersion. Next, the syringe pump was turned on, and fluorescence-labeled whole human blood was pipetted into the blood reservoir. The image recording process did not start until blood reached the perfusion chamber.

## 3. Results and Discussion

### 3.1. Surface Analysis

#### 3.1.1. X-ray Photoelectron Spectroscopy 

Angle-resolved XPS with take-off angles of 20 and 90 degrees relative to the surface was performed on mirror- then electrochemically polished 10 mm × 10 mm × 2 mm 316L stainless steel samples after each surface modification step. The principal objective of the investigation was to confirm the effectiveness of surface cleaning and activation protocol with piranha solution, as well as to establish the successful attachment of MEG-TFA (and subsequent formation of the MEG-OH adlayer) on steel samples. Elemental identification and modes characterized through deconvolution of carbon and oxygen narrow scan spectra were assigned based on the reported binding energy value obtained from the literature [45].

A layer of superficial contaminant is present on the bare sample surface despite the extensive mirror and electrochemical polishing efforts, as highlighted by the elemental C (1s) and Si (1s) relative atomic percentages at both 20 degrees and 90 degrees take-off angles (Table 1). Through the deconvolution of the O (1s) narrow scan spectrum abundant metal oxides and metal hydroxide species, primarily air-exposed Cr_2_O_3_ and Cr(OH)_3_, are evident respectively at 530.1 eV and 531.7 eV (Figure 2; additional high-resolution XP spectra are recorded in Appendix A). This observation is in agreement with previous knowledge that the generation of a chromium oxide layer on stainless steel surface blocks oxygen diffusion to the surface, thus contributing to its anti-corrosion properties. Following activation of the sample surface, the relative atomic percentages of elemental C (1s) at both 20-degree and 90-degree take-off angles decreases significantly, although there is evidence of adventitious C. As expected, the signal due to Si is removed, whereas those for Fe, Ni, and Mo, which are not previously found on the untreated bare sample, become detectable at both take-off angles (Table 2).

The appearance of an organic fluorine peak for MEG-TFA-modified steel represents convincing evidence that at least the trifluoroacetyl group is present on the surface. An intense and broad peak due to organic silicon (2p), which is located at ~103 eV, and not observed on activated sample surface, also appears at both take-off angles. The formation of the MEG-TFA adlayer on the substrate surface is further supported by the deconvoluted C (1s) narrow scan spectra at both take-off angles, where peaks due to CF3 and C = O moieties are now apparent at ~294 eV and 289 eV, respectively. Hydrolysis of the TFA group to produce MEG-OH yields spectra at both angles that confirm the removal of the F(1s) peak. A careful inspection of the deconvoluted C (1s) and O (1s) narrow scan spectra also reveals component peak heights and areas associated with C–O and C–C/C–H increase marginally, suggesting that the MEG backbone stayed intact and undamaged when TFA solvolysis was taking place (Table 2).

#### 3.1.2. Contact Angle Goniometry

Untreated bare stainless steel shows a hydrophobic surface with a contact angle (CA) of 86.6 ± 2.8° (*n* = 5) (Figure 3a). Following a rigorous surface-cleaning procedure and treatment with piranha solution, the activated substrate surface wettability significantly increases, as the CA drops to 48.9 ± 2.7° (*n* = 5), suggesting the surface has become hydrophilic and is fully saturated with hydroxyl groups. The contact angle of the MEG-TFA-modified surface returns to roughly the same level as the value for the bare sample, CA = 84.3 ± 4.6° (*n* = 5) due to the presence of hydrophobic TFA groups. After removal of the TFA group through solvolysis, the distal hydroxyl group of the MEG backbone is exposed, resulting in a substantial CA decrease to 19.7 ± 2.5° (*n* = 5). Overall, the contact angle data match the conclusions reached from the XPS surface characterization study detailed above, and agree with previously published work on MEG-OH [38]. Alongside the previously presented XPS data, this strongly suggests that 316L stainless steel was successfully modified with a MEG-OH surface coating.

### 3.2. Thrombogenicity Study of the Blood–Surface Interaction

#### 3.2.1. End-Stage Measurement of Surface Platelet Adhesion

Platelet aggregation on bare and MEG-OH surface-modified stainless steel following 20 min of interaction with whole human blood was studied for wall shear rates of 100 s^−1^, 300 s^−1^, 600 s^−1^, 750 s^−1^, and 900 s^−1^ employed during such exposure. The dye DiOC_6_ (3,3′-dihexyloxacarbocyanine iodide) chosen for this work is a commonly used cell-permeant, lipophilic, green fluorescent molecule primarily used for staining membrane-bound cellular organelles, such as mitochondria and endoplasmic reticulum [46]. Although not conventionally used for the labeling of platelets, the choice was made in this case for three reasons. First, DiOC_6_ does not cause platelet activation, even for extended time periods, and secondly, in the presence of red blood cells only platelets and leukocytes are fluorescence-labeled since hemoglobin within red blood cells effectively quenches any fluorescence [46]. Furthermore, the substantial size difference between leukocytes and platelets, ~12–17 µm versus ~1–2 µm in diameter, respectively, allows the facile distinction between the two particles under visualization by a high-resolution confocal microscope. Finally, platelet loading with DiOC_6_ is very straightforward, only requiring a minimum of 10 min of incubation at room temperature.

Representative frames for the bare and MEG-OH-modified surfaces as extracted after 20 min of blood perfusion at 100 s^−1^ and 600 s^−1^ shear rates are displayed in Figure 4 and Figure 5. From these frames, it is visually evident that cleaned bare steel surfaces, which were analyzed as the control group in this study, triggered a significant amount of platelet adhesion, aggregation, and thrombus formation after 20 min of exposure to fluorescence-labeled whole human blood. Platelet adhesion on MEG-OH-modified surfaces, highlighted by the red arrows in the frames, is barely observable.

Briefly, in order to examine the data on a quantitative basis, the frames were subjected first to a conversion to grayscale, followed by binarization. The binarized images were then imported to image processing software. The number of black (representing the background) and white (representing the fluorescent platelet) pixels in each selected frame can be calculated. The ratio of white pixels to the total pixel number in a thresholded frame, which can be calculated by the “area fraction”, is reflective of the surface coverage area associated with platelets. The surface platelet percentage coverage on the control groups was calculated at 99% confidence to be 7.72 ± 0.38% (*n* = 5), 5.79 ± 1.39% (*n* = 5), 4.98 ± 0.60% (*n* = 5), 1.08 ± 0.08% (*n* = 5), and 0.94 ± 0.11% (*n* = 5), for the 100 s^−1^, 300 s^−1^, 600 s^−1^, 750 s^−1^, and 900 s^−1^ shear rates, respectively (Figure 6). In sharp contrast, the MEG-OH-modified steel surfaces demonstrated remarkably superior resistance to thrombus formation after an identical period of exposure to blood (Figure 6). In this case, the calculated platelet percentage coverage on modified surfaces reported at 99% confidence interval are 0.67 ± 0.06% (*n* = 5) at 100 s^−1^, 0.29 ± 0.10% (*n* = 5) at 300 s^−1^, 0.37 ± 0.04% (*n* = 5) at 600 s^−1^, 0.10 ± 0.02% (*n* = 5) at 750 s^−1^, and 0.08 ± 0.01% at 900 s^−1^ shear rates, respectively. Student’s *t*-tests, assuming unequal sample variances, were performed to evaluate whether the calculated mean platelet surface percentage coverage on the bare and MEG-OH-modified steel surfaces is equal. The evaluated *t*-values for each blood shear rate are given in Table 3. The calculated Student’s *t*-values at each tested shear rate are much greater than the tabulated Student’s *t*-critical values at degrees of freedom = 4, which is reported as 4.604. Therefore, there is a highly statistically significant difference between platelet surface coverage on bare and MEG-OH-modified steel.

This reduction in platelet surface coverage was far greater than the amount of reduction observed for recent heparin drug-eluting stents [23,24,26], which showed far more platelet adhesion to their modified steel samples as compared to MEG-OH-coated samples, and poorer reduction in adhesion of less than 80% compared to the greater-than-90% adhesion observed for MEG-OH. Additionally, the amount of platelet reduction was greater than that observed for recent graphene surface coatings on steel which display an 80–90% reduction [30,31,32], and performed similarly to hydrogel coatings, with a greater-than-90% reduction [33,34]. As such, MEG-OH performs as well or better than the so-called state-of-the-art alternative coatings as it pertains to the prevention of thrombus formation.

MEG-OH is also low-cost and easy to apply to steel samples, which will make industrial application of this coating easier. This suggests that MEG-OH may perform better than these technologies if used on implanted stents, although further work on the long-term stability and biocompatibility of MEG-OH is needed. Our previous work has found MEG-OH coatings to be stable for multiple days [26,47], but further work is needed to assess longer timespans for modified steel samples.

#### 3.2.2. Real-Time Measurement of Surface Platelet Adhesion

The objective of this work was to assess the real-time, anti-thrombogenic profile of MEG-OH-modified steel surfaces compared to the bare version for extended periods of time. Although major changes were not made to the experimental protocol, it was necessary to modify a few parameters. In this study, the time allowed for fluorescence-labeled whole human blood perfusion was increased to 60 min. Therefore, a significantly larger volume of whole human blood was required for each perfusion. A much higher blood concentration of DiOC_6_ fluorescent dye also became mandatory in order to avoid complete photobleaching of labeled platelets already adhered to the substrate surface. In this situation, photobleached platelets would become unobservable under confocal microscope. In addition, the intensity of the excitation laser was turned down to ~50% of the intensity set for the previous experiment. Prior to each experiment, all perfusion chambers involved were thoroughly checked for air and liquid leakage to ensure blood can maintain laminar flow under constant shear rate for at least 60 min. One of each bare and MEG-OH-modified steel surfaces was tested under 100 s^−1^, 600 s^−1^, and 900 s^−1^ shear rates.

In this real-time surface thrombogenicity study, images for each prepared surface were recorded at 120 ms time intervals between frames, while the exposure time allowed for individual frame recording by the camera was set to 100 ms. After 60 min of blood perfusion at a specific shear rate over one surface was completed, individual frames captured at every 3 min were extracted from the image stack and processed using the same procedure as in the end-stage experiments. Twenty frames for every surface were recorded at each shear rate, starting from 3 min to 60 min perfusion time, with 3 min intervals between each frame.

The results of these experiments for the 100 and 600 s^−1^ rates are summarized in Figure 7 and Figure 8. Notably, the percentage coverage due to platelet adhesion and eventual thrombus formation on bare steel follows a near exponential increase pattern under the 100 s^−1^ shear rate, reaching an alarming ~53% surface coverage after 60 min of blood perfusion. Indeed, it was observed that the blood stream flow path becomes visibly obstructed by massive surface thrombus formation under 100 s^−1^ shear rate after 60 min. As the shear rate increases to 600 s^−1^ and 900 s^−1^ (not shown) platelet surface coverage on bare steel correspondingly decreases for each recorded time period. The platelet surface coverage in these cases nearly triples from 20 min to 60 min, eventually reaching 20.2% and 10.7%, respectively, for each of these two shear rates. In stark contrast, MEG-OH-modified steel surfaces clearly display a remarkably high degree of resistance to surface plate adhesion, evidenced by the reduction in surface platelet coverage by over 90% under all tested shear rates after 60 min of blood perfusion, compared with the bare samples. In addition, the surface platelet micro-aggregate formation on MEG-OH-surface-modified steel is observed to maintain a minimal and steady rate, particularly under the 600 s^−1^ and 900 s^−1^ shear rates. Shear rates in arteries are typically greater than 300 s^−1^ [48], which bodes well for MEG-OH coatings which perform better at higher shear rates. This suggests that the natural flow of blood, combined with a MEG-OH coating on an implanted stent, will greatly reduce the amount of thrombus formation for patients, reducing complications from implanted stents.

### 3.3. Basis of Reduction of Platelet Adhesion by MEG-OH Surface Modification

The success exhibited in this work with respect to the significant reduction in platelet adhesion to steel by surface-attached MEG-OH requires explanation. The generation of thrombi on foreign surfaces from blood is a complex process which is considered to first involve protein adsorption followed by platelet attachment [6]. Accordingly, it is apparent that the modified surface in this case is responsible for the curtailment of such a process. It has been known for some time that surfaces modified by polyethylene glycol films reduce protein adsorption via the instigation of a water “barrier”, the so-called kosmotropic effect. A more detailed view of the chemistry lies in an appreciation of the nature of water present at the 0.60 nm layer. Interstitial water in the MEG-OH monolayer is present as clusters which are tightly bound with limited lability and mobility, an effect which is instigated by the chain-internal -OH group [41]. Furthermore, it is important to note that in this condition, any release of water from such a hydrated adlayer during protein adsorption would clearly constitute a thermodynamic penalty. Adlayer flexibility, however—as a gauge of compressibility—is expected to play a secondary role. An alternative explanation is that the particular molecular structure of the MEG-OH surface configuration does not allow for interaction with protein groups by either hydrophobic or electrostatic forces, given that the distal OH group may well be unavailable for interaction.

## 4. Conclusions

The research described herein demonstrates that it is feasible to transfer oligoethylene silane surface chemistry, previously established for polymers, to 316L stainless steel, a widely used material for a number of biomedical implants, including stents. Surface characterization by XPS and contact angle goniometry has confirmed the successful modification of steel samples with MEG-OH coatings. Collectively, surface thrombogenicity assays involving fluorescence-labeled whole human blood has confirmed dramatic resistance to surface platelet adhesion and thrombus formation by the MEG-OH adlayer under all experimental shear rates. Statistical testing has indicated that there is a highly significant difference between platelet surface coverage on bare and MEG-OH-modified steel. The monolayer yields a reduction in surface platelet coverage by at least 90% on average, under all tested shear rates, in both end-stage and real-time blood perfusion experiments. In terms of relevance to stent technology, the oligoethylene monolayer provides a number of useful features in addition to its capability to reduce platelet aggregation. It is highly stable to standard sterilization protocols and can be stored without degradation for considerable periods of time, while also being low-cost, and simple to apply to samples. These advantages make it highly appealing for use on medical devices made of steel.

Despite the attractiveness of the silane chemistry evaluated here, a number of aspects warrant further investigation. For example, high platelet concentration can cause blood to be more sensitive to coagulation triggers. In order to avoid potential systemic biases and errors, it would be beneficial to study blood samples from a number of sources in future investigations. Finally, with respect to clinical application, long-term surface stability in blood can be crucial. There is an indication that certain organosilane adlayers are susceptible to cleavage by hydrolysis when immersed in aqueous or buffer environment for extended periods of time, especially at higher or lower pH conditions [49]. In this particular study, however, the presence of the critical and protective interstitial water layer instigated by the ethylene glycol moiety of MEG-OH discussed above was not involved for the surface silanes examined by the authors. Moreover, the highly retained antithrombogenic behavior of MEG-OH-modified polymers is well-established for experiments conducted with surfaces exposed to whole blood for several days [39].

## Figures and Tables

**Figure 1 materials-14-02342-f001:**
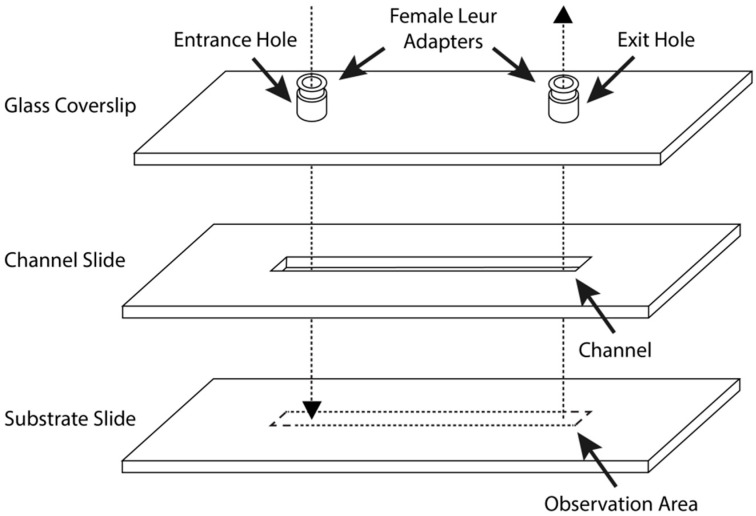
Schematic diagram of custom-made parallel-plate flow chamber.

**Figure 2 materials-14-02342-f002:**
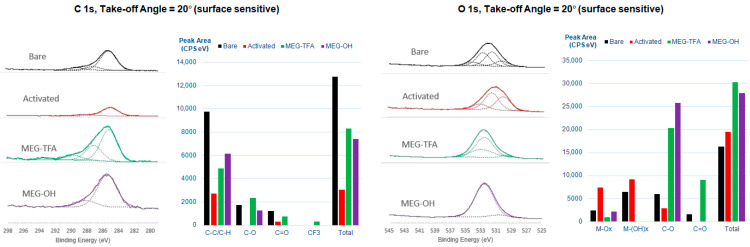
High-resolution XP spectra for carbon and oxygen scans of stainless steel through the stages of modification taken at a 20° take-off angle, additionally showing deconvolution of each signal.

**Figure 3 materials-14-02342-f003:**
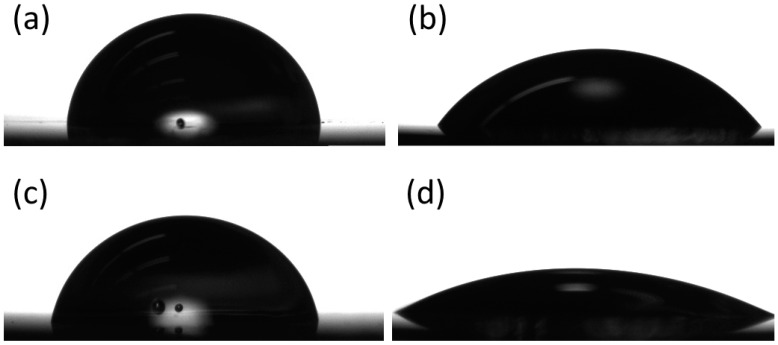
Example images of contact angle measurements for (**a**) bare stainless steel, (**b**) cleaned stainless steel, (**c**) MEG-TFA-modified stainless steel, and (**d**) MEG-OH-coated stainless steel.

**Figure 4 materials-14-02342-f004:**
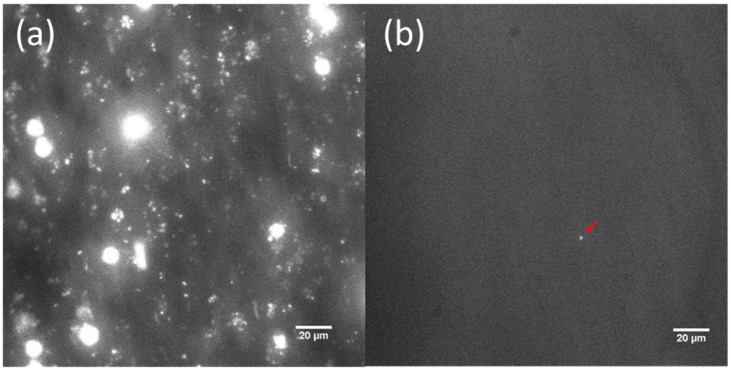
Representative images after 20 min of blood perfusion at 100 s^−1^ shear rate showing platelet aggregation on bare (**a**) and MEG-OH-modified 316L stainless steel (**b**).

**Figure 5 materials-14-02342-f005:**
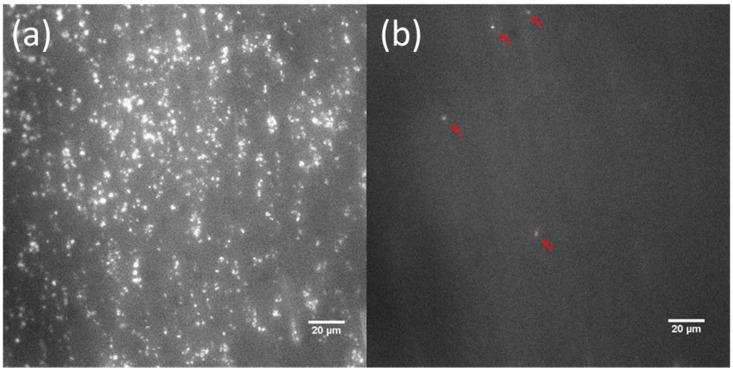
Representative images after 20 min blood perfusion at 600 s^−1^ shear rate showing platelet aggregation on bare (**a**) and MEG-OH-modified 316L stainless steel (**b**).

**Figure 6 materials-14-02342-f006:**
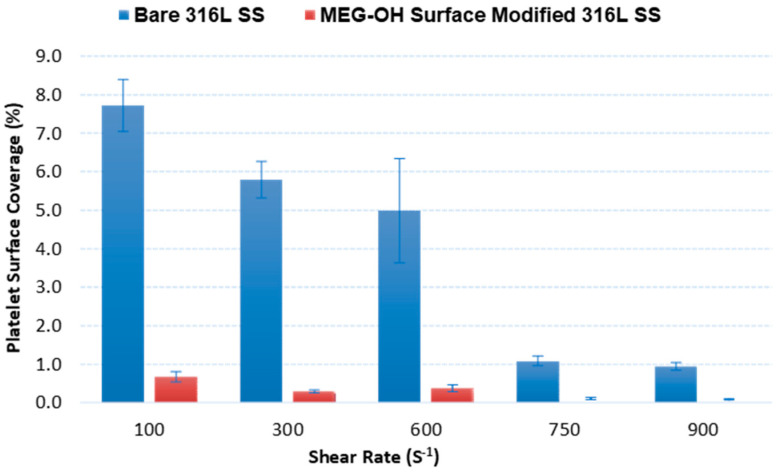
Percentage of surface coverage due to platelet adhesion on bare and MEG-OH modified 316L stainless steel after 20 min of blood perfusion at 100 s^−1^, 300 s^−1^, 600 s^−1^, 750 s^−1^, and 900 s^−1^ shear rates.

**Figure 7 materials-14-02342-f007:**
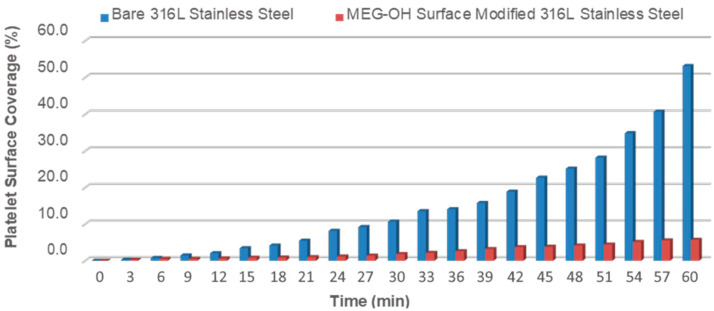
Real-time platelet surface percentage coverage on bare and MEG-OH-modified 316L stainless steel during 60 min of blood perfusion at 100 s^−1^ shear rate.

**Figure 8 materials-14-02342-f008:**
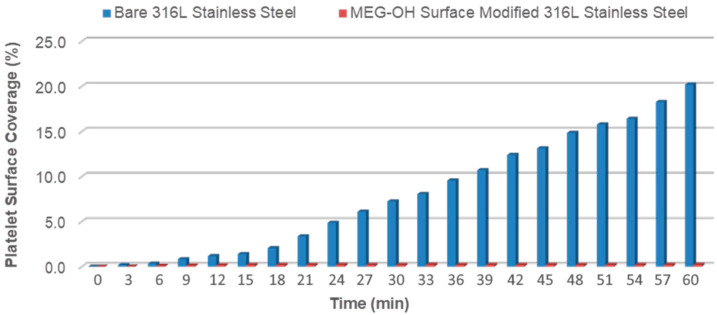
Real-time platelet surface percentage coverage on bare and MEG-OH-modified 316L stainless steel during 60 min blood perfusion at 600 s^−1^ shear rate.

**Table 1 materials-14-02342-t001:** Calculated volumetric flow rate and required blood volume at 100 s^−1^, 300 s^−1^, 600 s^−1^, 750 s^−1^, and 900 s^−1^ shear rates ^1^.

Shear Rate (s^−1^)	Blood Flow Rate (μL min^−1^)	Minimum Blood Volume Required (mL)	
		20 min	60 min
100	10.3	0.21	0.62
300	31	0.62	1.86
600	61.3	1.24	3.72
750	77.4	1.55	4.65
900	92.9	1.86	5.58

^1^ Calculated via *y* = 100Q/wh^2^ where *y*, Q, w, and h stand for wall shear rate (s^−1^), volumetric blood flow rate (mL/min), channel width (mm), and channel height (mm), respectively [44].

**Table 2 materials-14-02342-t002:** XPS relative atomic percentages for untreated (bare), oxidized (activated), MEG-TFA-modified (MEG-TFA) and MEG-OH-modified (MEG-OH) stainless steel at 20- and 90-degree take-off angles (TOA).

Sample	TOA	C (1s)	O (1s)	Si (1s)	F (1s)	Cr (2p)	Fe (2p)	Ni (2p)	Mo (2p)
Bare	20°	64.1	28.7	6.4	0	0.8	0	0	0
	90°	37.8	42.6	12.5	0	3.5	3.5	0	0
Activated	20°	11.6	58.6	0	0	15.1	14.8	0	0
	90°	9.7	59.3	0	0	10.1	8.2	6.4	6.4
MEG-TFA	20°	33.2	44.5	17.2	4.3	0.8	0	0	0
	90°	27.3	45.3	13.3	12.6	1.6	0	0	0
MEG-OH	20°	35.7	47.9	15.2	0	1.2	0	0	0
	90°	29.6	53.1	14.2	0	1.9	0.9	0	0.3

**Table 3 materials-14-02342-t003:** Calculated Student’s t-values for comparison of mean platelet surface percentage coverage on bare and MEG-OH-modified steel surfaces.

Shear Rate (s^−1^)	100	300	600	750	900
Student’s *t*-value	52.7	23.0	20.9	33.5	26.1
at 99% confidence					

## Data Availability

Data sharing is not applicable to this article.

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
