# Peer review of "Anti-Thrombogenicity Study of a Covalently-Attached Monolayer on Stent-Grade Stainless Steel"

_materials, 2021, doi:10.3390/ma14092342_

Round 1

Reviewer 1 Report

In the current paper, authors presented an attempt develop an ultrathin covalently-attached monolayer based on monoethylene glycol silane chemistry on 316L austenitic stainless steel to anti-thrombogenic in exposure of human blood. The paper is really great practical importance but lacks scientific analysis and discussion. Significance more information regards a practical notes for methodological research with statistical results than scientific analysis of obtained results. General comments that should be included to improve the value of paper:

Abstract and Introduction section:

First three sentences in the abstract should be integrated with the introduction section where the knowledge gap has been presented but reviewer is not sure what's new/ground-breaking in this work is presented? Is this approach is new and novel and to what extent? What makes authors work valuable in comparison to other work already published. There is no clear summary of introduction section.

Results and discussion section:

1. Results obtained in this section and each subsection separately should be supported by sufficient references better discussed, especially when ‘results’ and ‘discussion’ sections are combined into one. There are only some references used in this section without significant discussion and the obtained results were not compared or supported by similar works. I feel that too much experimental work was reported and not much significant discussed.

2. The XP spectra showing characteristic peaks from X-ray photoelectrons, both define the elements in the surface layer and also provide a lot of extremely important information about the state of the surface and phenomena occurring on it. Therefore, high resolution XP spectra should be constitute a better described documentation of the obtained results in the publication, and not be recorded in the Supplementary Materials.

Conclusion

Conclusions should be improve. Notifications about "Despite the attractiveness of the silane chemistry evaluated here, a number of aspects warrant further investigation..." up to the end of conclusion do not constitute results of the investigation in the manuscript, and could be include on the end Results and Discussion.

The paper is interesting at some point but I strongly suggest to provide more scientific discussion. Right now it looks like purely technical paper with very good methodology.

Author Response

Done by file attachment

Reviewer 2 Report

The paper can be considered for publication, but major revision is compulsory:

  1. Introduction must to be rewrittten since are not considering the publications from 2017 until now, with only one exception, position 26 at references. The paper must to be updated !!!
  2. A lot of references are not mentioned in the text of the paper: please reconsider the list and the state-of-the-art!
  3. Table 1: some data are missing at 60 min ...
  4. Figure 5 and 6: labels are moved away from the graphbars ...
  5. Englsi is very poor! A native English must to check and modify many sentences!!!

Author Response

Done by file attachment

Reviewer 3 Report

The study is interesting but needs to be revised and improved to be published in this journal. A number of suggestions are given below:

  1. In the introduction, the last paragraph should be devoted to the authors' goals and the conclusion should be omitted
  2. The methods section is very long while the results section is short
  3. In the results section, no references are used for discussion
  4. In the results of hydrophobicity analysis, use the image to analyze the results
  5. Use microscopic analysis (AFM, SEM, …) to analyze surface structure
  6. Can the results of this study be generalized to other stainless steels (316, 304, 321) ?
  7. It is recommended that you use the most up-to-date references in the materials journal

Author Response

Done by file attachment

Round 2

Reviewer 1 Report

The changes introduced in the abstract are apparent. The abstract still contains well-known and described in the literature general information regarding the possibility of 316L austenitic steel using in medicine, what could be obviously a regular information in Introduction. In the abstract should be note the research an ultrathin covalently-attached monolayer based on mono-ethylene glycol silane, which was deposited on the 316L steel substrate material, and it is the properties of this coating should constitute the information in the abstract based on obtained research results. The authors did not in any way investigate the 316L steel properties for coronary angioplasty applications such us structural, mechanical, and corrosion resistance, therefore it shouldn't be the subject in the Abstract at all.

The authors added 15 new literature in the Introduction, supplementing some literature information without detailing the results obtained. Potentially, this information can be considered sufficient for further study by the reader. However, both in the Introduction and in the all sections of manuscript, the numbering of references from [22] to [34], e.i. [22-34] is incorect after inserting 15 new references. The same problem concerns the incorrect numbering of Figures from 2 to 8 (Figs. 2-8) in the description of the research results in pharagraph 3 (Result & Discussion). Figure 2 is scanned with a very bad resolution, and in the XPS results analysis only new information is the insertion of Figure 2 without any supplementing, even elementary information about the surface condition and phenomena occurring on it.

Improved manuscript is still good methodology work and I believe that at the moment the paper is only technical notes for methodological research with statistical results than scientific analysis of obtained results.

The SEM / EDS results on the cross-section of the as-coated 316L steel substrate in the surface area should be also analyzed.

In the summary section, the effect of the obtained results constitutes only approx. 38% of the information, while the remaining content concerns the discussion on future works. The paper are not the standards typical of scientific publications in the Materials.

Reviewer 2 Report

It can be published as it is.

Reviewer 3 Report

The authors have modified the article well. Thanks to the respected authors. It is suggested that this article be accepted.
